# Secondary Degeneration of Oligodendrocyte Precursor Cells Occurs as Early as 24 h after Optic Nerve Injury in Rats

**DOI:** 10.3390/ijms24043463

**Published:** 2023-02-09

**Authors:** Lillian M. Toomey, Melissa G. Papini, Thomas O. Clarke, Alexander J. Wright, Eleanor Denham, Andrew Warnock, Terry McGonigle, Carole A. Bartlett, Melinda Fitzgerald, Chidozie C. Anyaegbu

**Affiliations:** 1Curtin Health Innovation Research Institute, Curtin University, Bentley, WA 6102, Australia; 2Perron Institute for Neurological and Translational Science, Sarich Neuroscience Research Institute Building, 8 Verdun St., Nedlands, WA 6009, Australia; 3Experimental and Regenerative Neurosciences, School of Biological Sciences, The University of Western Australia, Perth, WA 6009, Australia

**Keywords:** oligodendrocyte precursor cells, secondary degeneration, oxidative stress, DNA damage, proliferation, blood-brain barrier, CNS injury, optic nerve injury

## Abstract

Optic nerve injury causes secondary degeneration, a sequela that spreads damage from the primary injury to adjacent tissue, through mechanisms such as oxidative stress, apoptosis, and blood-brain barrier (BBB) dysfunction. Oligodendrocyte precursor cells (OPCs), a key component of the BBB and oligodendrogenesis, are vulnerable to oxidative deoxyribonucleic acid (DNA) damage by 3 days post-injury. However, it is unclear whether oxidative damage in OPCs occurs earlier at 1 day post-injury, or whether a critical ‘window-of-opportunity’ exists for therapeutic intervention. Here, a partial optic nerve transection rat model of secondary degeneration was used with immunohistochemistry to assess BBB dysfunction, oxidative stress, and proliferation in OPCs vulnerable to secondary degeneration. At 1 day post-injury, BBB breach and oxidative DNA damage were observed, alongside increased density of DNA-damaged proliferating cells. DNA-damaged cells underwent apoptosis (cleaved caspase3+), and apoptosis was associated with BBB breach. OPCs experienced DNA damage and apoptosis and were the major proliferating cell type with DNA damage. However, the majority of caspase3+ cells were not OPCs. These results provide novel insights into acute secondary degeneration mechanisms in the optic nerve, highlighting the need to consider early oxidative damage to OPCs in therapeutic efforts to limit degeneration following optic nerve injury.

## 1. Introduction

Injury to the central nervous system (CNS) involves two components of damage: the initial mechanical insult and a subsequent cascade of spreading damage called secondary degeneration [1]. The initial primary injury typically manifests as axonal shearing, contusions, and hemorrhage or hematoma [1,2]. Secondary degeneration occurs as pathological factors released by injured neurons and glia spread to the surrounding tissue, causing additional self-propagating damage [3]. Glaucoma and other optic neuropathies follow similar sequelae, where axonal injury to retinal ganglion cells triggers a plethora of degenerative mechanisms that spread throughout the visual system to progressively impair visual function [4]. Hypoxia-induced vascular dysfunction can reduce blood flow to the optic nerve head, contributing to axonal injury and vision loss in optic neuropathies [5,6]. The BBB, comprising endothelial cells, neurons, astrocytes, pericytes, and OPCs, is a key component of vascular function [7].

The rodent partial optic nerve transection model is especially useful for characterizing mechanisms of secondary damage in the optic nerve, facilitating spatial segregation between primary and secondary injury. Only the dorsal aspect of the right optic nerve is partially transected thus leaving the ventral aspect vulnerable solely to secondary degenerative processes [8]. Within the initially spared tissue, a multitude of secondary degeneration mechanisms occur, including oxidative stress, BBB dysfunction, reactive gliosis, axonal damage, dysmyelination, and oligodendrocyte death, associated with functional deficits post-injury [9,10]. Such secondary damage can spread as far into the brain as the superior colliculus via the visual pathways, to elicit a remote degenerative response following optic nerve injury [11].

A key feature of secondary degeneration that exacerbates acute pathology is oxidative stress. Oxidative stress occurs when the rate of reactive oxygen species production increases and overwhelms the detoxification capacity of the antioxidant system [12]. Excessive levels of reactive oxygen species can cause damage to a variety of cellular structures, including lipids, proteins, and DNA [13]. One form of oxidative DNA damage is nucleobase modifications, such as the guanine oxidation by-product 8-hydroxy-2-deoxyguanosine (8OHDG). Nucleobase modifications can cause particularly harmful effects by either modifying the genetic code or blocking DNA replication [14].

Within 5 min of a partial optic nerve transection, astrocytes are oxidatively stressed and become hypertrophic by 3 h [15]. No changes in Olig2+ oligodendroglia are observed by 24 h. However, at day 3, oligodendroglia, including OPCs, are significantly more likely to become oxidatively DNA damaged than other cell types within the ventral optic nerve [16]. The heightened vulnerability of OPCs is likely due in part to increased concentrations of Ca^2+^-permeable P2X_7_ receptors and α-amino-3-hydroxy-5-methyl-4-isoxazolepropionic acid (AMPA) receptors [17,18], increased intracellular iron levels [19], and decreased concentrations of antioxidant defenses [19,20]. Similarly, a specific subpopulation of newly derived mature oligodendrocytes has been identified to have elevated oxidative DNA damage compared to their pre-existing counterparts at this same time point [16]. The newly derived oligodendrocyte subpopulation was also less likely to become apoptotic than pre-existing oligodendrocytes, but instead demonstrated a decreased long-term capacity for myelination [16]. The concurrent death of pre-existing oligodendrocytes alongside the reduced myelination capacity of newly derived, DNA-damaged oligodendrocytes likely contributes to chronic deficits in myelination observed at 1, 3, and 6 months after optic nerve injury [21,22,23]. However, the specific vulnerability of oligodendroglia to this oxidative DNA damage at the earlier 1 day timepoint, is yet to be investigated. It is possible that a therapeutic window exists prior to 3 days, where oligodendroglia show signs of oxidative damage but are not yet proliferative or apoptotic.

To address this knowledge gap, this study assessed the vulnerability of OPCs to oxidative DNA damage at an acute 1 day timepoint following a partial optic nerve transection. Within the ventral nerve susceptible to secondary degeneration, 8OHDG DNA damage increased and correlated to the extent of BBB dysfunction. DNA-damaged cells showed increased proliferation with injury and increased apoptosis. While OPCs accounted for the majority of proliferating, DNA-damaged cells, these were not the largest population of apoptotic cells.

## 2. Results

### 2.1. Secondary Degeneration in the Ventral Optic Nerve following Partial Injury

At 1 day following a partial optic nerve transection, the ventral nerve was immunohistochemically assessed for 8OHDG, immunoglobulin G (IgG) and 5′Ethynyl-2-deoxyuridine (EdU) to quantify oxidative DNA damage, BBB dysfunction and cellular proliferation respectively. There was a significant effect of injury on both the area (t(4.448) = 17, *p* = 0.0004, Figure 1A) and mean intensity (t(4.197) = 17, *p* = 0.0006, Figure 1B) of 8OHDG immunoreactivity relative to sham controls, indicating increased oxidative DNA damage post-injury (Figure 1C). Similarly, the mean area of IgG immunoreactivity significantly increased with injury compared to sham controls (Mdn_sham_ = 2.018, Mdn_injured_ = 0.00825, U = 0, *p* < 0.0001, Figure 1D,F). The observed increase in IgG extravasation indicates an injury-induced breach of the BBB. A strong and significant positive monotonic relationship between the mean areas of IgG and 8OHDG immunoreactivities was also observed (r_s_ = 0.752, *p* = 0.0003, Figure 1E), suggesting that animals with increased DNA damage in the ventral optic nerve typically experience greater levels of BBB dysfunction.

The effect of partial optic nerve transection on cellular proliferation as indicated by EdU+ staining in the ventral nerve was also assessed. There was a trend towards an increased density of EdU+ cells with injury compared to sham controls, although this difference did not reach statistical significance (t(2.072) = 17, *p* = 0.0538, Figure 2A). EdU+ cells were then categorized based on colocalization with 8OHDG above a set threshold and a two-way ANOVA was used to compare the densities of proliferating cells either with or without oxidative DNA damage (F(1,34) = 2.947, Figure 2B,D). A significant difference was observed by Tukey *post-hoc* comparisons in the density of EdU+ 8OHDG+ cells with injury compared to the sham group (*p* = 0.0264). There were no differences in the density of EdU+ 8OHDG− cells with injury compared to those without injury (*p* > 0.05). A weak but significant positive monotonic relationship between the mean area of IgG immunointensity and EdU+ densities was also observed (r_s_ = 0.489, *p* = 0.0394, Figure 2C).

Additionally, the effect of partial optic nerve transection on apoptosis was assessed by detecting Cleaved Caspase3, the proteolytically-cleaved and functionally-active form of Caspase3 [24]. A significant increase was observed in the overall density of Cleaved Caspase3+ cells with injury compared to sham controls in the ventral nerve (t(5.064) = 16, *p* = 0.0001, Figure 3A). Cleaved Caspase3+ cells were then categorized based on colocalization with 8OHDG above a set threshold and a two-way ANOVA was used to compare the densities of apoptotic cells either with or without oxidative DNA damage (F(1,32) = 25.64, Figure 3B,D). A significant difference was observed by Tukey post hoc comparisons in the density of Cleaved Caspase3+ 8OHDG+ cells with injury compared to Cleaved Caspase3+ 8OHDG+ cells in the sham group (*p* < 0.0001). A significant increase was also found between Cleaved Caspase3+ cells with and without 8OHDG+ in both the injured (*p* < 0.0001) and sham groups (*p* < 0.0001). No differences were observed in the density of Cleaved Caspase3+ 8OHDG− cells with injury compared to Cleaved Caspase3+ 8OHDG− cells without injury (*p* > 0.05). A moderate and significant positive monotonic relationship between the mean area of IgG immunointensity and Cleaved Caspase3+ densities was also observed (r_s_ = 0.694, *p* = 0.003, Figure 3C).

### 2.2. Heightened Vulnerability of OPCs to Oxidative DNA Damage

To identify oxidatively damaged OPCs, antibodies detecting neural/glial antigen 2 (NG2) [25] or platelet-derived growth factor receptor α (PDGFRα) [26] were utilized. Within NG2+ glia specifically, the mean intensity of 8OHDG immunoreactivity significantly increased compared to sham controls (t(2.161) = 15, *p* = 0.0472, Figure 4A,B). Correspondingly, there was a significant increase with injury in the mean intensity of 8OHDG within PDGFRα+ glia (t(3.992) = 17, *p* = 0.0007, Figure 4C,D).

### 2.3. Proliferative and Apoptotic Status of OPCs with Oxidative DNA Damage

The proliferative status of oxidatively DNA-damaged OPCs was then assessed within the ventral nerve. The EdU+ population was first identified as either PDGFRα+ or PDGFRα− cells and analyzed using a two-way ANOVA (F(1,36) = 1.457, Figure 5A). Tukey post hoc comparisons revealed a significant increase in the density of EdU+ PDGFRα+ OPCs with injury compared to sham (*p* = 0.0466). No significant difference was observed in the density of EdU+ PDGFRα− cells with injury (*p* > 0.05). The EdU+ PDGFRα+ and EdU+ PDGFRα− populations were further classified by colocalization with 8OHDG DNA damage and analyzed via a three-way ANOVA with Tukey post hoc comparisons (F(1, 64) = 1.958, Figure 5B,C). There was a significant increase with injury in the density of EdU+ PDGFRα+ 8OHDG+ cells compared to sham (*p* = 0.0115). Interestingly, the densities of EdU+ PDGFRα+ 8OHDG+ cells were also significantly higher than EdU+ PDGFRα+ 8OHDG− cells within injured animals (*p* = 0.0033). No significant differences were observed with injury for EdU+ PDGFRα+ 8OHDG− OPCs (*p* > 0.05), or EdU+ PDGFRα− 8OHDG+ (*p* > 0.05) or EdU+ PDGFRα− 8OHDG− (*p* > 0.05) cells compared to sham. Altogether, this suggests that OPCs are the major proliferating, DNA-damaged cell type at 1 day following injury to the optic nerve.

The apoptotic status of oxidatively DNA-damaged OPCs was also assessed within the ventral nerve. The Cleaved Caspase3+ population was initially identified as either PDGFRα+ or PDGFRα− cells and analyzed using a two-way ANOVA (F(1,32) = 5.611, Figure 6A). Tukey post hoc comparisons revealed no significant increase in the density of Cleaved Caspase3+ PDGFRα+ OPCs with injury compared to sham (*p* > 0.05). A significant difference was observed in the density of Cleaved Caspase3+ PDGFRα− cells with injury (*p* < 0.0001). There was also a significant increase in the density of Cleaved Caspase3+ PDGFRα− cells compared to Cleaved Caspase3+ PDGFRα+ cells within both injured (*p* < 0.0001) and sham groups (*p* < 0.0001). The Cleaved Caspase3+ PDGFRα+ and Cleaved Caspase3+ PDGFRα− populations were further classified by colocalization with 8OHDG DNA damage and analyzed via a three-way ANOVA with Tukey *post-hoc* comparisons (F(1, 64) = 33.37, Figure 6B,C). There was a significant increase with injury in the density of both Cleaved Caspase3+ PDGFRα+ 8OHDG+ OPCs (*p* = 0.0304) and Cleaved Caspase3+ PDGFRα− 8OHDG+ cells (*p* < 0.0001) compared to sham. Within the sham group, there were significantly more Cleaved Caspase3+ PDGFRα− 8OHDG+ cells than Cleaved Caspase3+ PDGFRα+ 8OHDG+ OPCs (*p* < 0.0001), Cleaved Caspase3+ PDGFRα+ 8OHDG− OPCs (*p* < 0.0001) and Cleaved Caspase3+ PDGFRα− 8OHDG− cells (*p* < 0.0001). Similarly, within the injured group, there were significantly more Cleaved Caspase3+ PDGFRα− 8OHDG+ cells than OPCs that were Cleaved Caspase3+ PDGFRα+ 8OHDG+ (*p* < 0.0001) or Cleaved Caspase3+ PDGFRα+ 8OHDG− (*p* < 0.0001) or Cleaved Caspase3+ PDGFRα− 8OHDG− cells (*p* < 0.0001). In addition, there was a significant increase in the density of Cleaved Caspase3+ PDGFRα+ 8OHDG+ OPCs compared to both Cleaved Caspase3+ PDGFRα+ 8OHDG− OPCs (*p* = 0.0019) and Cleaved Caspase3+ PDGFRα− 8OHDG− cells (*p* = 0.0019) within the injured group. These data indicate an increase in apoptosis of DNA-damaged OPCs at 1 day following injury to the optic nerve, though these cells do not form the majority of the overall apoptotic cell population.

## 3. Discussion

This study investigated the role of oxidative DNA damage to OPCs following optic nerve injury at an acute 1 day timepoint. Early pathological changes induced within the ventral nerve were indicative of secondary degeneration mechanisms, with observed increases in BBB dysfunction and 8OHDG DNA damage, as well as increased cellular proliferation specifically in DNA-damaged cells. Heightened DNA damage was also specifically identified within both NG2+ glia and PDGFRα+ glia, indicating a vulnerability of OPCs to oxidative DNA damage. Apoptotic cells were DNA damaged, and associated with BBB breach. Finally, this study demonstrated that while the PDGFRα+ OPC population was the major proliferating, DNA-damaged cell type following injury to the optic nerve, most of the apoptotic cells were not OPCs. While additional research is needed to further delineate the role of oxidative damage post-injury, these novel results provide valuable insights into early mechanisms that underpin secondary degeneration of the optic nerve at 1 day following injury.

In line with previous work that assessed outcomes at 1 day following a partial optic nerve transection, this study found significant increases in DNA damage and BBB dysfunction in the ventral optic nerve vulnerable to secondary degeneration [10,27]. A strong and significant relationship between the extent of oxidative DNA damage and BBB breach was also uncovered and observed across animals in both sham and injured groups. Though oxidative stress had already been closely associated with BBB dysfunction in a variety of CNS diseases and injuries [28], a direct relationship between the two has not been previously investigated within the partial optic nerve transection model. Taken together with the increased oxidative damage to OPCs, this direct relationship suggests that the observed increase in parenchymal IgG relates to a breakdown of the OPC component of the BBB and not due to transcellular transport [29]. The density of DNA-damaged cells undergoing proliferation significantly increased with injury. Meanwhile, there was no change in the density of proliferating cells without DNA damage as indicated by 8OHDG immunoreactivity. This finding suggests a relationship between oxidative DNA damage and cellular proliferation that requires further elucidation. Additionally, it will be important to determine whether oxidative damage is driving, or is a consequence of, secondary pathological mechanisms.

Consistent with previous studies [25,30], this study used NG2 and PDGFRα separately to identify OPCs. In our hands, immunohistochemical detection reliably allowed quantification of one cell identifying marker together with the functional indicators 8OHDG and Cleaved Caspase3. We, therefore, identified OPCs using a combination of their expression of NG2 or PDGFRα and known morphology (i.e., round features with small processes; in line with several other studies [25,31]). The majority of the PDGFRα+ cells in this study were most likely OPCs, as OPCs have the most abundant expression of PDGFRα in the CNS [26] and PDGFRα is the best singular marker for OPCs [25]. PDGFRα and NG2 likely identified a similar population of OPCs in this study, as the distribution and morphology of PDGFRα+ cells coincide with NG2+ cells in the brain [30,32], and PDGFRα+ and NG2+ cells showed a similar pattern of oxidative damage.

OPCs have previously been shown to be vulnerable to DNA damage at 3 days following a partial optic nerve transection [16]. The present study showed that this pathology occurs as early as 1 day after injury, with increases in oxidative DNA damage specifically observed within both NG2+ glia and PDGFRα+ glia. The PDGFRα+ OPC population was the major proliferating and DNA-damaged cell type. This finding builds on previous work in this model which showed that approximately 54% of proliferating cells were NG2+ Olig2+ OPCs at 1 day post-injury [33]. This early proliferative response of OPCs to injury occurs prior to the onset of cell death at 7 days, with OPC loss continuing out to 3 months post-injury [33]. Therefore, the observed early proliferation does not prevent a chronic depletion of OPCs later in the pathological sequelae. Combined with the observed increase in the proliferation of OPCs with oxidative DNA damage specifically within the injured nerve, these data suggest that proliferation could be an early indicator of OPC damage and dysfunction following optic nerve injury. However, whether increased OPC proliferation is actively induced by DNA damage or whether already proliferating OPCs are inherently more vulnerable to oxidative stress mechanisms post-injury is not yet known. Indeed, not all of the cells that are proliferating post-injury are OPCs, with a variety of cells known to proliferate following CNS injury, including astrocytes and microglia [34,35]. Nevertheless, the data suggest that oxidatively damaged OPCs drive the majority of cellular proliferation acutely post-injury.

As OPCs differentiate into mature oligodendrocytes post-injury, a peak ratio of proliferating to non-proliferating OPCs occurs at 3 days post-injury before these EdU+ OPCs differentiate through the stages of the oligodendroglial lineage towards mature myelinating oligodendrocytes [16]. By 3 days following a partial optic nerve transection, there is also a specific subpopulation of newly derived mature oligodendrocytes that have increased levels of DNA damage compared to their pre-existing counterparts [16]. Therefore, it is highly likely that a proportion of the identified proliferating and DNA-damaged OPC population at 1 day post-injury may differentiate into mature oligodendrocytes at later timepoints. This subpopulation of DNA-damaged, newly derived mature oligodendrocytes are less likely to become apoptotic than pre-existing oligodendrocytes but demonstrate a decreased long-term capacity for myelination [16]. The decreased apoptosis of newly derived and proliferating oligodendrocytes suggests that proliferation and differentiation may have been protective against the cell death associated with DNA damage post-injury. Nevertheless, the concurrent death of pre-existing oligodendrocytes alongside the reduced myelination capacity of newly derived, DNA-damaged oligodendrocytes likely contributes to chronic deficits in myelination following injury.

Heterogeneity within the overall OPC population [36] could convey varying degrees of susceptibility to oxidative damage. For example, some OPCs colocalize and interact with blood vessels, whilst others reside solely in the brain parenchyma [37]. OPCs with an intermediate phenotype have also been observed, whereby they are both simultaneously perivascular and parenchymal [38], suggesting a spectrum of OPC phenotypes based on association with the vasculature. Functional differences between OPC subpopulations identified here remain to be investigated. However, it may be that a portion of the observed DNA-damaged OPCs modulated detrimental effects at the BBB. OPCs located at the vasculature have already been found to play a key role in BBB integrity under pathological conditions, such as cerebral hypoperfusion [39] and MS [40,41]. Therefore, future studies should determine the relative vulnerability of OPC subpopulations to oxidative DNA damage to elucidate any potential contribution of perivascular OPC damage in pathological BBB dysfunction. It is also important to note that OPCs do not exist in isolation, and cross-talk between OPCs and other cells, both in the parenchyma and at the perivascular regions, is likely to contribute to outcomes.

Female rats were used in this study to enable initial comparison with our previously published work and to address the disproportionate overrepresentation of male animals within neuroscience literature. Future studies will include both male and female rats to identify possible sex-dependent changes. However, it is noteworthy that differences were observed between injured and uninjured female rats, indicating that any potential effect of female hormones does not preclude the detection of injury-induced changes in these animals.

This study identified OPCs as the major proliferating, DNA-damaged cells acutely following optic nerve injury. The observed early oxidative damage to this cell type likely plays a key role in exacerbating pathology post-injury, further highlighting oxidative stress as a therapeutic target worthy of future investigation. Pharmacological modulation of glial components of the BBB, such as the aquaporin-4 water channel on astrocytes, has been shown to reduce vasogenic edema and improve function in rats with CNS injury [42]. Given the importance of OPCs for BBB and CNS function, therapeutic interventions that attenuate oxidative DNA damage in OPCs are likely to mitigate the progression of secondary degeneration to axonal and functional loss after injury. Computer-aided, high-throughput drug screening platforms that investigate up to a hundred thousand compounds per day have the potential to accelerate the discovery or repurposing of drugs that effectively target oxidative stress-induced OPC dysfunction [43]. High-resolution imaging of OPCs and associated cells in humanized, self-organized 3D organoids or microvessel-on-a-chip platforms would facilitate the robust assessment of drug candidates likely to be effective in humans [44,45].

## 4. Materials and Methods

### 4.1. Animal Procedures and Study Design

Twenty adult, female PVG rats (180 g) were obtained from the Animal Resource Centre in Murdoch, Western Australia. All procedures were in accordance with the principles of the National Health and Medical Research Council (NHMRC) of Australia Code of Practice for use of Animals for Scientific Purposes and were approved by the Animal Ethics Committee of The University of Western Australia (RA/3/100/1485) and the Animal Ethics Committee of Curtin University (ARE2017-4). The rats were provided ad libitum access to both food and water and were housed under a 12 h light/dark cycle. Rats were also given a 1 week acclimatization period to the holding facility prior to commencing the experimental period. The cohort consisted of two experimental groups: a sham control group (*n* = 10) and an injured group (*n* = 10).

### 4.2. Surgical Procedures

Partial optic nerve transections were performed as previously described [8], under anesthesia with intraperitoneal Ketamine (Ketamil, 50 mg/kg, Troy Laboratories, Glendenning, Australia) and Xylazine (Ilium Xylazil, 10 mg/kg, Troy Laboratories). In brief, the right optic nerve was surgically exposed about 1 mm behind the eye and the dorsal aspect of the nerve was partially lesioned to approximately 200 μm using a diamond radial keratotomy knife (Geuder, Heidelberg, Germany). Rats that underwent a sham injury received all surgical procedures except the cut in the surrounding nerve sheath and the partial transection into the optic nerve. Post-operative analgesia (Carprofen, 2.8 mg/kg, Norbook, Newry, UK) and sterile phosphate-buffered saline (PBS, 1 mL) were administered subcutaneously following surgery. To label cells actively undergoing the cell cycle, EdU (20 mg/kg, Invitrogen, Waltham, MA, USA) was delivered via intraperitoneal injection twice, once immediately following the surgical procedures during post-operative care and once the following morning at least 2 h prior to euthanasia. The total number of sham control animals was reduced to *n* = 9 due to *n* = 1 rat being resistant to the anesthesia necessary for surgery and thus omitted from the study. There were no deaths from the surgical procedure.

### 4.3. Tissue Processing

At 1 day post-injury, rats were euthanized with pentobarbitone sodium (160 mg/kg, Delvet) prior to being transcardially perfused with 0.9% saline followed by 4% paraformaldehyde (Sigma-Aldrich, St. Louis, MO, USA). The injured right optic nerves were dissected and immersed in a 4% paraformaldehyde solution overnight. The following day, the nerves were transferred to 5% sucrose (ChemSupply Australia, Bedford, South Australia, Australia), 0.1% sodium azide (Sigma-Aldrich) in PBS for cryoprotection. The optic nerves were then cryosectioned transversely at 14 µm, collected onto Superfrost Plus glass microscope slides, and stored at −80 °C prior to immunohistochemical analysis.

### 4.4. Multiplex Immunohistochemistry

Prior to commencing immunohistochemical analysis, the back surface of each slide was placed unsubmerged in PBS within an electrophoresis tank for 1 h at 70 V to reinforce the electrostatic bond between the tissue and slide, mitigating the risk of tissue detachment during the multiple wash steps involved in the protocol. Slides were then dried in a 37 °C oven for 10 min. Antigen retrieval involved heating the slides in 10 mM Tris-EDTA-0.5 M NaCl (pH 9.0) solution in the microwave for 2 min and 20 s, followed by a 20 min cooling period at room temperature. Slides were washed in a PBS bath prior to the application of a hydrophobic barrier around the tissue, using a PAP pen (Merck, Advanced PAP Pen, Darmstadt, Germany). Sections were washed with PBS three times before incubation with Peroxidazed 1 solution (0.3% H_2_O_2,_ PX968M, 121219-2, Biocare Medical, Pacheco, CA, USA) for 10 min at room temperature. Slides were washed three times in PBS after the application period for each reagent was complete. Non-specific background was blocked using 3% bovine serum albumin solution (Merck, 12657) for 20 min at room temperature.

Primary antibodies used recognized: 8OHDG (1:250, 4 µg/mL, mouse, Abcam, ab62623, GR3284216-13), rat Immunoglobulin G (IgG, 1:150, 10 µg/mL, goat, BA-9400, ZG0108, Vector Laboratories, Burlingame, CA, USA), NG2 (1:50, 20 µg/mL, rabbit, AB5320B, 3218879, Merck), Cleaved Caspase3 (Asp175, 1:150; rabbit, D3E9, #9579, Lot 1, Cell Signaling Technology, Danvers, MA, USA) and PDGFRα (1:250, 4 µg/mL, rabbit, PA516571, VJ2870528A, ThermoFisher, Waltham, MA, USA). Primary antibodies were diluted in PBS and applied overnight at 4 °C. Target markers were detected sequentially in three separate combinations of antibodies/detection systems—Combination 1: PDGFRα, 8OHDG, EdU; Combination 2: PDGFRα, 8OHDG, Cleaved Caspase3; and Combination 3: NG2, 8OHDG. A separate section was used for each combination. IgG was detected alone on a separate section.

Fluorescence labeling of PDGFRα, 8OHDG, and Cleaved Caspase3 was performed using the following secondary antibodies, respectively: AF488-conjugated anti-rabbit IgG antibody (1:400; 5 μg/mL, donkey, A21206, 2289872, ThermoFisher), AF647-conjugated anti-mouse IgG antibody (1:100; 20 μg/mL, donkey, A31571, 2136787, ThermoFisher) and AF555-conjugated anti-rabbit IgG antibody (1:400; 5 μg/mL, donkey, A31572, 1945911, ThermoFisher). Secondary antibodies were diluted in PBS and applied for 2 h at room temperature. To minimize cross-reactivity between the two anti-rabbit secondary antibodies for each combination, anti-rabbit IgG (H+L) antibody (1:100, 15 μg/mL, horse, BA-1100-1.5, ZH0421, Vector Laboratories) was applied for 1 h at room temperature after the full detection of the first rabbit antibody (i.e., after primary and secondary antibody application steps). The anti-rabbit IgG (H+L) antibody saturates the remaining binding sites for rabbit-specific secondary antibodies on the Fc region of the preceding rabbit primary antibody, restricting off-target binding when the next rabbit-specific secondary antibody is applied. The biotinylated NG2 and IgG antibodies were fluorescently labeled using the VECTASTAIN^®^ Elite^®^ ABC-HRP Kit (1:100, PK-6100, ZG0312, Vector Laboratories) in conjunction with a TSA FLUORESCEIN REAGENT PACK (NEL701A001KT, 191230019, Akoya Biosciences, Marlborough, MA, USA) according to manufacturers’ instructions. To detect EdU+ cells, the Click-iT EdU AlexaFluor-647 Imaging Kit (C10340, 2284610, ThermoFisher) was utilized according to the manufacturer’s instructions. Finally, sections were washed with PBS three times and coverslipped using Fluoromount-G (Thermo Fisher).

### 4.5. Imaging and Analysis

For each analysis, the entire optic nerve was visualized using a Nikon A1 confocal microscope (Nikon Corporation, Sydney, Australia) or a Dragonfly High Speed Confocal Microscope System (Andor Technology, Belfast, UK). A series of images were taken at 0.5 μm increments along the z-axis with consistent capture settings across all images for each outcome, at a magnification of 20× and numerical aperture of 0.75. Image analysis was performed using Fiji/ImageJ image processing software (National Institutes of Health, Bethesda, MD, USA).

The area of the ventral nerve region was segmented and quantified. Representative immunointensity thresholds for each outcome measure were determined to distinguish positive signals from the background prior to analyses within the ventral nerve. Using the most in-focus visual z-slice and the defined intensity thresholds, the mean area and intensity of immunoreactivity for IgG and 8OHDG were then semi-quantified. The areas above threshold measurements were normalized to the total area of the ventral nerve region. The number of EdU+ and Cleaved Caspase3+ cells was counted within the ventral nerve, normalized against the total ventral area, and expressed as the mean number of cells/mm^2^. Both EdU+ 8OHDG+ and Cleaved Caspase3+ 8OHDG+ cells were detected by the colocalization of either EdU+ or Cleaved Caspase3+ cells with 8OHDG immunointensity above the set threshold similarly quantified.

Using the defined thresholds to identify NG2+ glia and PDGFRα+ glia, the intensity of 8OHDG was then also measured within the identified glia to assess the levels of DNA damage specifically within these cell types. EdU+ and Cleaved Caspase3+ cells were categorized into PDGFRα+ and PDGFRα− subpopulations based on immunoreactivity and cellular morphology. These subpopulations were further categorized via the colocalized detection of 8OHDG DNA damage.

### 4.6. Statistics

The obtained data were analyzed and plotted using GraphPad PRISM 9 software. All outcome measures, except for the area of IgG immunointensity, satisfied the assumption of normality according to a Kolmogorov–Smirnov test. Therefore, a t-test, two-way ANOVA with Tukey’s post hoc or three-way ANOVA with Tukey’s post hoc were used as appropriate. Given the area of IgG immunointensity did not satisfy the assumption of normality, the non-parametric Mann–Whitney test was used to assess the statistical difference between sham and injury for this outcome measure. A Spearman’s correlation was used to assess the monotonic relationship between IgG and either 8OHDG, EdU or Cleaved Caspase3. Any reductions in final n’s reflect a loss of tissue from slides during immunohistochemical analyses or the exclusion of sections that had become damaged during tissue processing and analysis in a way that precluded reliable quantification of outcomes. Statistical significances shown on graphs are hypothesis-driven and may not display all significant differences obtained. Specifically, only significant differences in comparable cell types between the sham control and injured groups are shown, as well as any differences found between cells within each group. No data outliers were removed for any outcome measures.

## Figures and Tables

**Figure 1 ijms-24-03463-f001:**
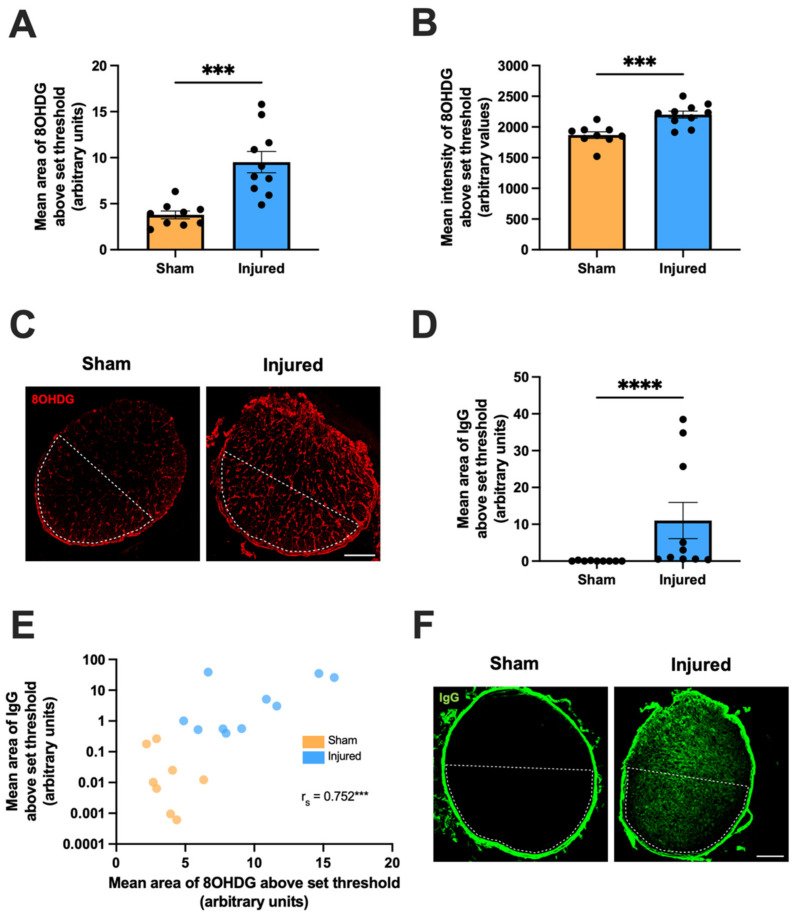
Effect of injury on oxidative DNA damage and BBB dysfunction within the ventral optic nerve, relative to sham injury. Area (**A**) and mean intensity (**B**) of 8OHDG immunoreactivity within the ventral optic nerve were assessed to determine the level of oxidative DNA damage. Graphs display individual data points overlaid on a bar displaying the mean ± SEM. *n* = 9–10 rats per group. Statistical analysis by *t*-tests. (**D**) Area of IgG immunointensity within the ventral nerve was assessed to determine the extent of BBB breach. Graph displays individual data points overlaid on a bar displaying the mean ± SEM. Statistical analysis by Mann-Whitney test. *n* = 9–10 rats per group. (**E**) The area of IgG immunointensity was correlated to the area of 8OHDG using Spearman’s correlation, with the r_s_ value and corresponding *p*-value displayed on the graph. The mean area of IgG immunointensity was plotted on a log scale to best illustrate the overall relationship between IgG and 8OHDG on the scatterplot. Each data point on the graph represents an individual animal. *n* = 8–10 rats per group. (**C**,**F**) Representative images of both 8OHDG and IgG immunoreactivity in sham and injured rats are shown, scale bars = 100 μm. Area of the ventral nerve is denoted by dotted lines. All areas above threshold measurements are presented in arbitrary units as the data have been normalized to the total area of the ventral nerve for each animal. No outliers were removed for any outcome measure. Significant differences are indicated by *** *p* ≤ 0.001, **** *p* ≤ 0.0001.

**Figure 2 ijms-24-03463-f002:**
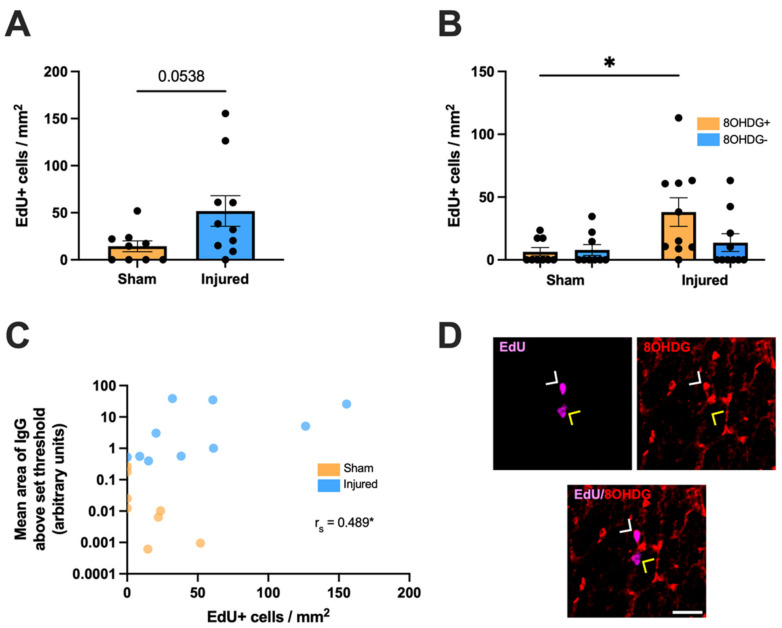
Effect of injury on cellular proliferation within the ventral optic nerve. (**A**) The density of EdU+ cells was quantified in the ventral optic nerve following partial optic nerve transection or sham injury. Graphs display individual data points overlaid on a bar displaying the mean ± SEM. *n* = 9–10 rats per group. Statistical analysis by *t*-test. (**B**) The relative densities of EdU+ 8OHDG+ and EdU+ 8OHDG− cells were quantified. Statistical analysis by two-way ANOVA and Tukey *post-hoc* tests. Graphs display individual data points overlaid on a bar displaying the mean ± SEM. *n* = 9–10 rats per group. (**C**) The area of IgG immunointensity was correlated to the density of EdU+ cells using Spearman’s correlation, with the r_s_ value and corresponding *p*-value displayed on the graph. The mean area of IgG immunointensity was plotted on a log scale to best illustrate the overall relationship between IgG and EdU on the scatterplot. Each data point on the graph represents an individual animal. *n* = 8–10 rats per group. (**D**) Representative image of both an EdU+ 8OHDG+ cell (white arrow head) and an EdU+ 8OHDG− cell (yellow arrow head) is shown, scale bar = 25 μm. No outliers were removed for any outcome measure. Significant differences are indicated by * *p* ≤ 0.05.

**Figure 3 ijms-24-03463-f003:**
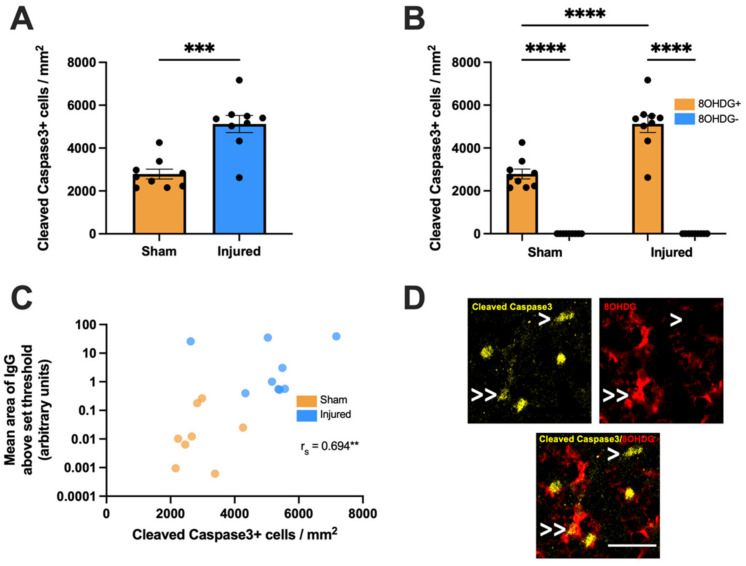
Effect of injury on apoptosis within the ventral optic nerve. (**A**) The density of Cleaved Caspase3+ cells was quantified in the ventral optic nerve following partial optic nerve transection or sham injury. Graphs display individual data points overlaid on a bar displaying the mean ± SEM. *n* = 9 rats per group. Statistical analysis by *t*-test. (**B**) The relative densities of Cleaved Caspase3+ 8OHDG+ and Cleaved Caspase3+ 8OHDG− cells were quantified. Statistical analysis by two-way ANOVA and Tukey post hoc tests. Graphs display individual data points overlaid on a bar displaying the mean ± SEM. *n* = 9 rats per group. (**C**) The area of IgG immunointensity was correlated to the density of Cleaved Caspase3+ cells using Spearman’s correlation, with the r_s_ value and corresponding *p*-value displayed on the graph. The mean area of IgG immunointensity was plotted on a log scale to best illustrate the overall relationship between IgG and Cleaved Caspase3+ on the scatterplot. Each data point on the graph represents an individual animal. *n* = 8–9 rats per group. (**D**) Representative image of both a Cleaved Caspase3+ 8OHDG+ cell (indicated by >>) and a Cleaved Caspase3+ 8OHDG− cell (indicated by >) is shown, scale bar = 25 μm. No outliers were removed for any outcome measure. Significant differences are indicated by ** *p* ≤ 0.01, *** *p* ≤ 0.001, **** *p* ≤ 0.0001.

**Figure 4 ijms-24-03463-f004:**
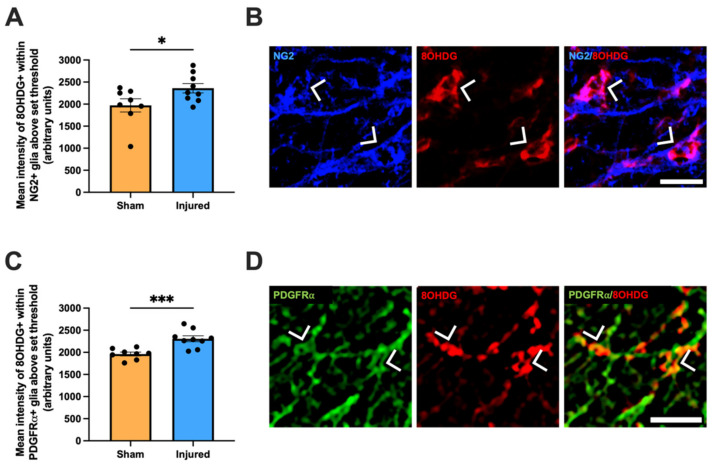
Effect of injury on oxidative DNA damage within NG2+ glia and PDGFRα+ glia. The mean immunointensity of 8OHDG was specifically quantified in NG2+ glial cells (**A**) and PDGFRα+ glia cells (**C**) in the ventral nerve. Graphs display individual data points overlaid on a bar displaying the mean ± SEM. *n* = 8–10 rats per group. Statistical analysis by *t*-tests. Representative images of NG2+ glia (**B**) and PDGFRα+ glia (**C**) with 8OHDG+ DNA damage are shown, indicated with arrow heads, scale bars = 20 μm. No outliers were removed for any outcome measure. Significant differences are indicated by * *p* ≤ 0.05, *** *p* ≤ 0.001.

**Figure 5 ijms-24-03463-f005:**
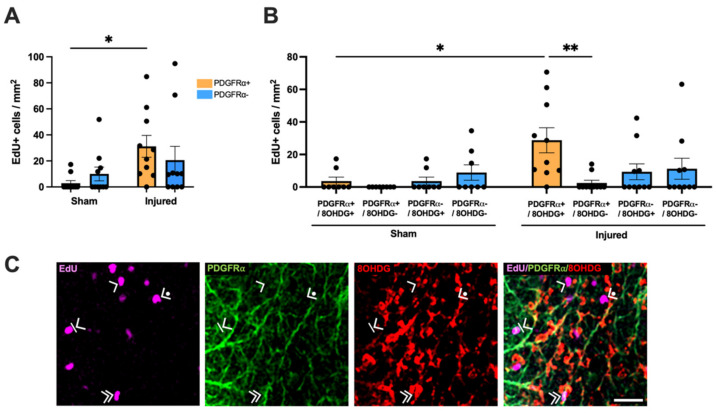
Effect of injury on proliferation and oxidative DNA damage within OPCs. (**A**) The relative densities of EdU+ PDGFRα+ and EdU+ PDGFRα− OPCs were quantified. Statistical analysis by two-way ANOVA and Tukey post hoc tests. (**B**) The relative densities of EdU+ cells colocalized with PDGFRα and 8OHDG were quantified. Statistical analysis by three-way ANOVA with Tukey post hoc tests. Graphs display individual data points overlaid on a bar displaying the mean ± SEM. *n* = 8–10 rats per group. (**C**) Representative images illustrating EdU+ PDGFRα+ 8OHDG+ OPCs (indicated by >>), EdU+ PDGFRα+ 8OHDG− OPCs (indicated by >|), EdU+ PDGFRα− 8OHDG+ cells (indicated by >), EdU+ PDGFRα− 8OHDG− cells (indicated by ●>) are shown, scale bar = 25 μm. No outliers were removed for any outcome measure. Significant differences are indicated by * *p* ≤ 0.05, ** *p* ≤ 0.01.

**Figure 6 ijms-24-03463-f006:**
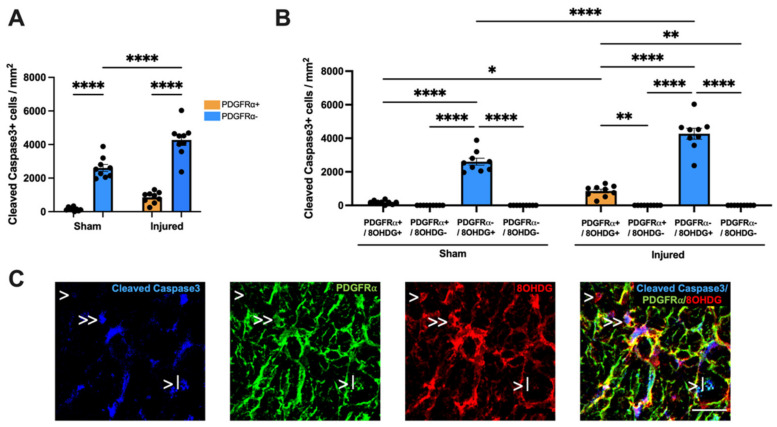
Effect of injury on apoptosis and oxidative DNA damage of OPCs. (**A**) The relative densities of Cleaved Caspase3+ PDGFRα+ and Cleaved Caspase3+ PDGFRα− OPCs were quantified. Statistical analysis by two-way ANOVA and Tukey post hoc tests. (**B**) The relative densities of Cleaved Caspase3+ cells colocalized with PDGFRα and 8OHDG were quantified. Statistical analysis by three-way ANOVA with Tukey post hoc tests. Graphs display individual data points overlaid on a bar displaying the mean ± SEM. *n* = 9 rats per group. (**C**) Representative image illustrating Cleaved Caspase3+ PDGFRα+ 8OHDG+ cells (indicated by >>), Cleaved Caspase3+ PDGFRα+ 8OHDG− cells (indicated by >|) and Cleaved Caspase3+ PDGFRα− 8OHDG+ cells (indicated by >) is shown. Cleaved Caspase3+ PDGFRα− 8OHDG− cells were not observed in the ventral nerve. Scale bar = 25μm. No outliers were removed for any outcome measure. Significant differences are indicated by * *p* ≤ 0.05, ** *p* ≤ 0.01, **** *p* ≤ 0.0001.

## Data Availability

The datasets generated during this study are available from the corresponding author upon request.

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
