# Peer review of "Secondary Degeneration of Oligodendrocyte Precursor Cells Occurs as Early as 24 h after Optic Nerve Injury in Rats"

_ijms, 2023, doi:10.3390/ijms24043463_

Round 1
Reviewer 1 Report
In this manuscript, the authors tried to figure out the early effects of oxidative damage to oligodendrocyte precursor cells(OPCs) in the acute optic injury in rats. They discovered that at 1 day postinjury, BBB disruption and oxidative DNA damage and increased proliferating cells were shown. They also found OPCs are the major proliferating cell type with DNA damage, but not the major capsase3+ cells. They provide a new view for acute secondary degeneration in the optic nerve, however, there are some points needed to be addressed.
1. Caspase3 is not a marker for apoptosis, the author should use cleaved-caspase3 for the indicator of apoptosis.
2. As the authors mentioned NG2 and PDGFRA can not only label OPCs but also label vascular cells, the best way to indicate OPCs is co-label of NG2 and PDGFRA or NG2 and Olig2 or PDGFRA and Olig2.
3. The title “Proliferating oligodendrocyte precursor cells are vulnerable to oxidative damage following acute optic nerve injury in rats” can mislead the readers, because the authors showed oxidative damage following acute optic nerve injury in rats increased the proliferation rate and apoptosis percentage, however, no evidence showed proliferating OPCs are vulnerable to oxidative damage while non-proliferating OPC are not.
Reviewer 2 Report
Optic nerve injury causes secondary degeneration through mechanisms such as oxidative stress, apoptosis and blood-brain barrier (BBB) dysfunction. In this work the authors made a partial optic nerve transection rat model of secondary degeneration and used immunohistochemistry to assess BBB dysfunction, oxidative stress and proliferation in oligodendrocyte precursor cells (OPCs) vulnerable to secondary degeneration at an acute 1 day timepoint. This is a good study. However, did not advance our knowledge about the cellular and molecular events caused by optical nerve injury. A severe study limitation was the very limited study period, just one day.
Reviewer 3 Report
Dear Editor,
The manuscript investigates the acute secondary degeneration mechanisms in the optic nerve, highlighting the need to consider early oxidative damage to OPCs in therapeutic efforts to limit secondary degeneration following optic nerve injury.
The design of the study and the technical quality of the work look convincing and results can be of general interest. However, there is a number of points that would need to be addressed in order to improve the quality of this paper before it can be accepted for publication:
Major:
-Animal work: the manuscript needs to include the mortality rate.
Response 1:
We have specified the mortality rate in Section 4.2 - Surgical Procedures
-This manuscript overlooked some essential and up-to-date work regarding the pathophysiology of BBB, for example:
· The introduction lacks the mention of BBB cell types.
Response 2a:
We have mentioned the BBB cell types in the first paragraph of the Introduction.
· Authors omitting a breakthrough study from 2020 by Kitchen et al in Cell, demonstrating that targeting glial cells reduces pressure in the CNS. That study shows pharmacological inhibition promotes functional recovery in injured rats. This role has been recently been confirmed by the work of Sylvain et al BBA 2021 using a photothrombotic stroke model. They have also shown a link to brain energy metabolism as indicated by the increase of glycogen levels.
- https://pubmed.ncbi.nlm.nih.gov/32413299/
https://pubmed.ncbi.nlm.nih.gov/33561476/
Response 2b:
We thank the reviewer for this suggestion. We have added the Kitchen et al study to the last paragraph of the Discussion to highlight the potential of pharmacological modulation of glial components of the BBB.
· Authors need to discuss the new classification of stages of BBB damage following CNS injuries and hypoxia. References:
https://academic.oup.com/brain/advance-article/doi/10.1093/brain/awab311/6367770
Response 2c:
The potential impact of hypoxia on vascular dysfunction has been added to the first paragraph of the Introduction. The BBB dysfunction observed in this study likely relates to a breakdown of the OPC component of the BBB and not due to transcellular transport, as there was a direct relationship between BBB dysfunction and oxidative damage, to which OPCs are vulnerable. This discussion has been added to the second paragraph of the Discussion.
· Oxidative damage to the CNS isn’t a disease of neurons only. Authors need to discuss recent trends in targeting the molecular and signalling mechanisms of astrocytes and other glial cells rather than just the traditional approaches. The importance of this new approach has been discussed in these references which should be included to enrich the discussion of current manuscript. References:
https://pubmed.ncbi.nlm.nih.gov/34973181/
https://pubmed.ncbi.nlm.nih.gov/34863533/
Response 3:
The references recommended by the reviewer discuss the molecular mechanisms governing pharmacological modulation of aquaporin-4. While we acknowledge the role of aquaporin-4 in vasogenic edema, we respectfully submit that this topic is more relevant to CNS edema and therefore beyond the scope/intent of our study, which focused on vulnerability of OPCs to secondary degeneration mechanisms.
-Hypoxia has been shown to affect proliferative and apoptotic pathways in glial cells. Hypoxia-induced molecular mechanisms on glial cells need to be discussed: References:
https://pubmed.ncbi.nlm.nih.gov/28925524/
https://pubmed.ncbi.nlm.nih.gov/29311824/
Response 4:
We thank the reviewer for highlighting the potential effect of hypoxia on glia behaviour. We have now mentioned the impact of hypoxia on vascular dysfunction in the first paragraph of the Introduction.
-Hypoxia has been shown to reduce BBB integrity and this allows the entry of more inflammatory mediators. Reference to be added:
https://www.ncbi.nlm.nih.gov/pmc/articles/PMC9798958/
Response 5:
In light of the reviewer’s suggestion, we have mentioned the impact of hypoxia on vascular dysfunction in the first paragraph of the Introduction. However, we did not include the suggested article as it discussed neuronal implications of cerebral malaria, which is beyond the scope of our study.
Minor:
Towards the end of discussion, authors need to discuss recent trends in targeting the molecular and signalling mechanisms of glial cells rather than just the traditional approaches. The importance of this new approach has been discussed in these references which should be included to enrich the discussion of current manuscript. References:
https://pubmed.ncbi.nlm.nih.gov/34973181/
https://www.mdpi.com/1422-0067/23/3/1388
Response 6:
We thank the reviewer for this suggestion. We have modified the last paragraph of the Discussion to highlight the potential of the targeting the molecular and signalling mechanisms of glia.
-CNS injuries are yet incurable diseases in most cases. Authors need to point out to the recent advances in applying the use of high-throughput screening and computer-aided drug design as have been nicely reviewed by Aldewachi et al 2021 as they can provide a novel insight that can support AQP target validation in future studies. References to be included:
https://pubmed.ncbi.nlm.nih.gov/33925236/
https://pubmed.ncbi.nlm.nih.gov/33672148/
Response 7:
We have modified the last paragraph of the Discussion to highlight the potential utility of computer‑aided drug design for effective development of treatments for neurodegenerative diseases.
- Conclusion and future prospects: Authors need to briefly discuss future directions following towards the end of their discussion and conclusion. This could include, but not limit to, the use of humanized self-organized models, organoids, 3D cultures and human microvessel-on-a-chip platforms especially those which are amenable for advanced imaging such as TEM and expansion microscopy since they enable real-time monitoring of inflammatory mediators following oxidative stress. References to be included:
https://pubmed.ncbi.nlm.nih.gov/32300301/
https://pubmed.ncbi.nlm.nih.gov/33117784/
Response 8:
We agree with the reviewer that high resolution imaging of OPCs and associated cells in humanized, self-organised 3D organoids or microvessel-on-a-chip platforms would facilitate robust assessment of drug candidates likely to be effective in humans. We have therefore mentioned this in the last paragraph of the Discussion.
Best.
Round 2
Reviewer 1 Report
The revised manuscript looks better now. The authors answers all my questions properly. There is only one minor point:
The authors changed caspase3 to cleaved-caspase3 in the context, but in the figures, they still used caspase3. Please use cleaved-caspase3 in the figure as well.
Reviewer 3 Report
Dear Editor,
The authors have successfully addressed the majority of my comments and concerns in order to improve the quality of the manuscript.
I believe that the new sections, improved ones, and updated references, have contributed to enhancing the clarity of the manuscript, which I can now endorse for publication.
All the best!
